

# Assessing kinesthetic proprioceptive function of the upper limb: a novel dynamic movement reproduction task using a robotic arm

Kristof Vandael[1,2], Tasha R. Stanton[3,4] and Ann Meulders[1,5]

[1] Experimental Health Psychology, University of Maastricht, Maastricht, Netherlands
[2] Laboratory of Biological Psychology, Katholieke Universiteit Leuven, Leuven, Belgium
[3] Neuroscience Research Australia, Randwick, New South Wales, Australia
[4] IIMPACT in Health, University of South Australia, Adelaide, South Australia, Australia
[5] Research Group Health Psychology, Katholieke Universiteit Leuven, Leuven, Belgium

## ABSTRACT

**Background:** Proprioception refers to the perception of motion and position of the body or body segments in space. A wide range of proprioceptive tests exists, although tests dynamically evaluating sensorimotor integration during upper limb movement are scarce. We introduce a novel task to evaluate kinesthetic proprioceptive function during complex upper limb movements using a robotic device. We aimed to evaluate the test–retest reliability of this newly developed Dynamic Movement Reproduction (DMR) task. Furthermore, we assessed reliability of the commonly used Joint Reposition (JR) task of the elbow, evaluated the association between both tasks, and explored the influence of visual information (viewing arm movement or not) on performance during both tasks.

**Methods:** During the DMR task, participants actively reproduced movement patterns while holding a handle attached to the robotic arm, with the device encoding actual position throughout movement. In the JR task, participants actively reproduced forearm positions; with the final arm position evaluated using an angle measurement tool. The difference between target movement pattern/position and reproduced movement pattern/position served as measures of accuracy. In study 1 ($N = 23$), pain-free participants performed both tasks at two test sessions, 24-h apart, both with and without visual information available (i.e., vision occluded using a blindfold). In study 2 ($N = 64$), an independent sample of pain-free participants performed the same tasks in a single session to replicate findings regarding the association between both tasks and the influence of visual information.

**Results:** The DMR task accuracy showed good-to-excellent test–retest reliability, while JR task reliability was poor: measurements did not remain sufficiently stable over testing days. The DMR and JR tasks were only weakly associated. Adding visual information (i.e., watching arm movement) had different performance effects on the tasks: it increased JR accuracy but decreased DMR accuracy, though only when the DMR task started with visual information available (i.e., an order effect).

**Discussion:** The DMR task's highly standardized protocol (i.e., largely automated), precise measurement and involvement of the entire upper limb kinetic chain (i.e., shoulder, elbow and wrist joints) make it a promising tool. Moreover, the poor association between the JR and DMR tasks indicates that they likely capture unique

Corresponding author
Ann Meulders,
ann.meulders@kuleuven.be

![PeerJ logo]

aspects of proprioceptive function. While the former mainly captures position sense, the latter appears to capture sensorimotor integration processes underlying kinesthesia, largely independent of position sense. Finally, our results show that the integration of visual and proprioceptive information is not straightforward: additional visual information of arm movement does not necessarily make active movement reproduction more accurate, on the contrary, when movement is complex, vision appears to make it worse.

## INTRODUCTION

When we perform controlled voluntary movements, such as reaching for a glass of water, we rely heavily upon sensory information elicited from the movement to successfully perform and control that movement. A key source of sensory information is proprioceptive input—it allows for the perception of motion and position of the body or body segments in space (*Proske & Gandevia, 2012*). Proprioceptive input consists of an ensemble of sensory information from various receptors that detect and encode the mechanical changes in tissues (e.g., muscles, skin) during movement. During active movement, muscle spindles are considered the primary source of proprioceptive information (*Proske & Gandevia, 2012*). Proprioceptive input then undergoes processing within the spinal cord, cephalad transmission up the sensory neuraxis, finally leading to a proprioceptive representation within the brain (i.e., area 2 of the primary somatosensory cortex in case of arm movement; *Chowdhury, Glaser & Miller, 2020*). During movement, proprioceptive (and tactile) input is used to inform motor planning (*Wolpert, Goodbody & Husain, 1998*). It is also used to determine whether or not the movement has occurred as intended—i.e., a motor efferent copy is generated and compared to the sensory input that has resulted as a consequence of this movement (*Wolpert & Ghahramani, 2000*; *Wolpert et al., 1995*). Such a process of sensorimotor integration ultimately allows for accurate, controlled movement.

A large variety of tests exist to quantify proprioceptive function, which differ in the required motor and memory capacity to perform the test, but importantly, also vary in the aspect of proprioception that they evaluate (*Hillier, Immink & Thewlis, 2015*). One aspect of proprioceptive function involves the perception of motion (i.e., kinesthesia), which is typically evaluated using a task in which the joint of interest is passively moved until the subject indicates they sense the movement and/or its direction (*Juul-Kristensen et al., 2008b*). Alternatively, to assess perception of spatial location or position (i.e., position sense), limb position reproduction tasks, such as the Joint Repositioning (JR) task are commonly used (*Han et al., 2016*). In the active variant, participants have their vision occluded and reproduce target *positions* using the body part of interest (e.g., using target positions of the forearm to assess position sense at the elbow joint). The average difference between target and reproduced position then serves as a measure of *accuracy*.

A limitation of these proprioceptive tasks is that they generally do not allow for evaluation of more complex processes that are essential for accurate and controlled movement (i.e., kinesthesia during functional movement), such as integration of sensory and motor information. This is an important limitation because goal-directed movement requires dynamic updating of motor output based on proprioceptively encoded (and changing) body position (i.e., sensorimotor integration; *Proske & Gandevia, 2012*). Evaluation of such processes underlying kinesthesia during active movement may provide unique and important information, given that there are known dynamic modulations that occur during movement (e.g., sensory gating; *Saradjian, 2015*). Capturing processes of dynamic modulation may also be important because proprioceptive tasks evaluating position sense or passively evaluating kinesthesia provide little insight into dynamic movement; that is, they are not always associated with actual motor performance (*Davies et al., 2006*; *Dukelow et al., 2012*; *Helsen et al., 2016*; *Kitchen & Miall, 2019*). Here we introduce a novel task in which movement *patterns* are reproduced to dynamically assess kinesthetic proprioceptive function: the Dynamic Movement Reproduction (DMR) task. Using the HapticMaster (Motekforce Link, Amsterdam, Netherlands), a three degrees-of-freedom, force-controlled robotic arm, this task involves continuous (i.e., online) assessment of an actively reproduced arm movement, thus including aspects of both limb position sense and sensorimotor integration to support kinesthetic function. The ability to accurately assess kinesthetic proprioceptive function during complex movement processes is clinically relevant given that a wide range of clinical conditions are characterized by impaired proprioception (*Goble, 2010*; *Proske & Gandevia, 2012*; *Röijezon, Clark & Treleaven, 2015*) and that the type of proprioceptive deficit can vary (*Kenzie et al., 2017*), meaning that it may be integration processes (vs. position sense) that are of crucial importance in certain clinical conditions.

A key feature for both research and clinical relevance of a proprioceptive function task is adequate test–retest reliability. Past work shows that the reliability of traditional, active JR tests ranges widely depending on the device used and the extremity joint measured (*Clark, Röijezon & Treleaven, 2015*; *Elangovan, Herrmann & Konczak, 2014*). Equipment measurement error likely influences these reliability findings. Use of more sophisticated equipment during testing, such as robotic devices (e.g., the HapticMaster)—which are becoming increasingly prevalent in research and clinical practice—may have higher sensitivity and precision (*Maggioni et al., 2016*). Such properties also affect the ability to detect proprioceptive impairment, which is essential given that even slight impairments might be of clinical relevance, particularly for complex sensorimotor integration processes. Therefore, the primary aims of the current study were to evaluate (1) test–retest reliability of the DMR task and (2) a JR test of the elbow, and (3) the association between performance on both tasks. Understanding the association between the tasks is important— if highly associated, then a complex task (such as the DMR task) might not be needed; if only weakly associated, then it would provide evidence that these tasks capture different aspects of proprioceptive function. Thus to evaluate these aims, in Study 1, healthy participants performed the DMR and JR tasks at two different test sessions, 24-h apart. Since the use of a robotic device allows for a highly standardized protocol and precise

measurement, i.e., features shown to increase test–retest reliability (*Maggioni et al., 2016*), we hypothesised that (1) the DMR task would be highly reliable (good-to-excellent range). Furthermore, consistent with findings from *Juul-Kristensen et al. (2008b)*, we hypothesised that (2) JR accuracy at the elbow would have fair-to-good reliability. Finally, we predicted (3) a weak association between DMR and JR accuracy. Both tasks involve active elbow movements and involve aspects of joint position sense; however, the continuous measurement of error during the DMR task likely also captures complex sensorimotor integration processes, thus only a weak association was anticipated.

In addition, to better understand the various sensory contributions to task performance (i.e., a task involving sensorimotor integration), our secondary aim was to evaluate the influence of visual information on both proprioceptive measures. Movements typically involve integration of visual and proprioceptive information, which may be combined in differing ways based on the nature of movement (e.g., differing between trajectory control and final position regulation; *Scheidt et al., 2005*). Testing both tasks with and without visual information of limb movement allows us to determine the relative visual vs. proprioceptive weighting in task performance. We hypothesised that there would be increased accuracy with visual information for both tasks, given that vision provides an extra source of sensory information that may assist in movement and joint position accuracy. Given that recent calls to improve research rigor recommend undertaking validation of study findings in an independent sample (*Laraway et al., 2019*), we also evaluated the two proprioceptive tasks (with and without visual information available) in a second independent sample (Study 2) to ensure reproducibility of our findings.

## STUDY 1: MATERIALS & METHODS

### Participants

Twenty-nine pain-free volunteers (sample size based on *Juul-Kristensen et al. (2008b)*) were recruited through word-of-mouth and the participant recruitment system of Maastricht University (Sona; Sona Systems, Nijmegen, The Netherlands). Six participants were excluded: four due to equipment failure and two because they confused movement directions (e.g., performing clockwise movements, when counterclockwise movements were requested). Statistical analyses were run on the final sample of $N = 23$ (mean (SD) age = 24.39 (3.12), ranging from 18–32, 11 women). Exclusion criteria were: chronic pain; left-handedness; uncorrected problems with hearing or vision; current pain at the dominant hand, wrist, elbow, or shoulder that may hinder task performance. Participants received €7.5 in gift vouchers as compensation for their time and effort.

### Pre-registration of the protocol and ethical approval

The experimental protocol was approved by the Ethics Review Committee Psychology and Neuroscience of Maastricht University (185 09 11 2017 S5) and pre-registered on AsPredicted (https://aspredicted.org/blind.php?x=c7g8bp). Prior to the start of the experiment, all participants read an information sheet, completed an exclusion criteria checklist, and provided written informed consent.

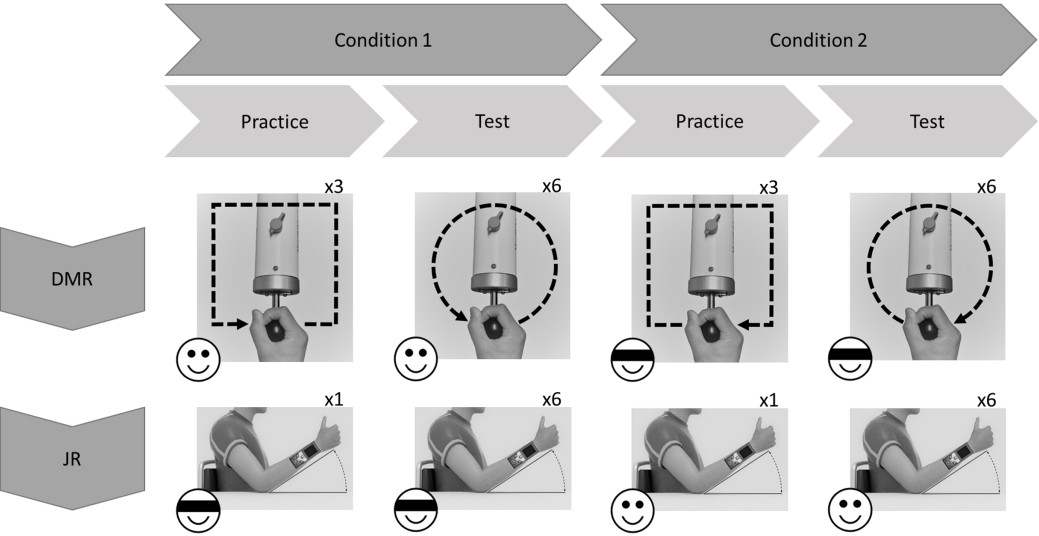

**Figure 1 Exemplary flowchart of a single test session.** Order of tasks and conditions (with or without visual information; indicated by emoji) was randomized across participants. In this example Dynamic Movement Reproduction (DMR) task is first, and Joint Reposition (JR) task is second. For DMR task, movement direction (clockwise or counterclockwise; indicated by arrows) was counterbalanced across conditions, and practice movements were squares, while test movements were circles, to minimize training effects.                                                                                                                         

## Study design

All participants performed both the JR and the DMR tasks using the dominant (right) arm. Each task comprised two conditions: Visual Information (i.e., without blindfold) and No Visual Information (i.e., with blindfold). In the Visual Information condition, participants directly watched the movement of their own limb. The order of the tasks and conditions was randomized across participants (using random.org). The same tasks were performed 24 h later, in the same order as during the first test session (Fig. 1).

## Apparatus

### Angle measurement tool (JR task)

The Bosch GLM 80 Professional measuring tool (Robert Bosch GmbH, Gerlingen-Schillerhoehe, Germany) was used to measure arm positions (in degrees; precision = $0.1°$; accuracy = $±0.2°$). The device was attached to participants' wrist using a Velcro strap and measured the angle of the forearm relative to the horizontal surface.

### HapticMaster (DMR task)

The HapticMaster (Motekforce Link, Amsterdam, The Netherlands; Fig. 2) is a 3 degrees-of-freedom force-controlled robotic arm. Participants hold the handle of the device and can move it in all directions within a specific volume of space.
The HapticMaster allows forward/backward movement with a depth of 40 cm, upward/downward movement with a height of 40 cm, and 60 degrees of rotation around its vertical axis (with smallest radius 46 cm; see youtube.com/watch?v=AQNNaEZ_Klg for a demonstration). In the current task, movements were confined to a 2-dimensional

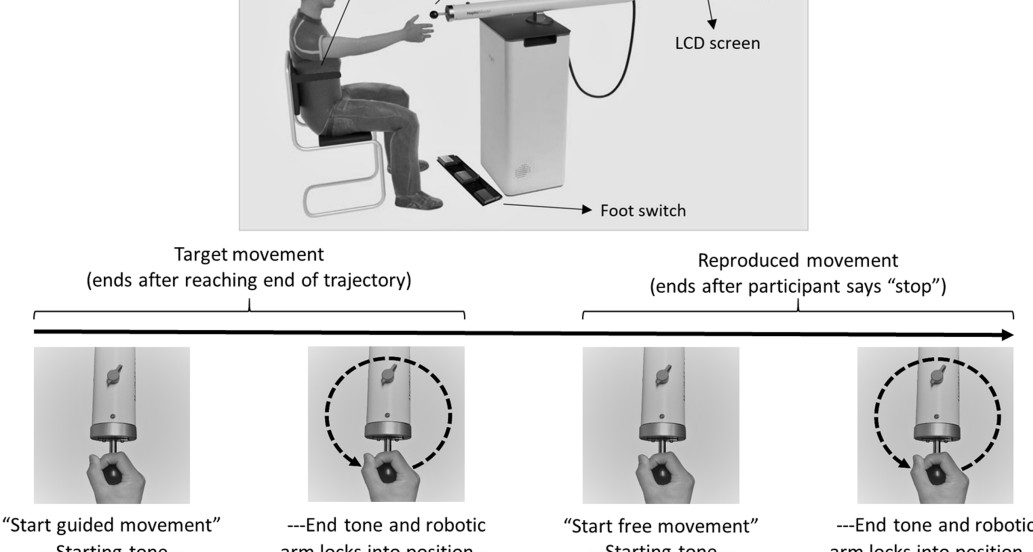

**Figure 2 Experimental setup and trial flow of Dynamic Movement Reproduction task.** Note that during target movements the robotic arm haptically delineated the trajectory, while during movement reproduction there was no guidance. On the right side of the participant, a partition (not displayed here) separated the experimenter and participant to prevent potential distractions during performance of the task.

horizontal movement plane (i.e., height remained constant). The robotic arm can be moved by exerting force on a sensor (i.e., circular handgrip) attached at the end of the arm. The HapticMaster automatically logs position along all three dimensions every 2 ms, with a resolution of $10^{-6}$ m.

### Computer software and hardware (DMR task)

The DMR task was programmed in C# using Unity 2017 (Unity Technologies, San Francisco, CA, USA). The experimental task was run on a Windows 10 Enterprise (Microsoft Corporation, Redmond, WA, USA) 64-bit Intel Core desktop computer (Intel Corporation, Santa Clara, CA, USA) and instructions were presented on a 40-inch LCD screen (Samsung UE40ES5500; Samsung Group, Seoul, South Korea). A Windows 10 compatible foot switch (USB Triple Foot Switch II; Scythe Co., Ltd., Tokyo, Japan) was used to navigate through instructions.

### Experimental setting

In both tasks, participants sat with their back against the chair and a strap encircling their torso to ensure their position remained fixed. For the DMR task, participants sat in front of the HapticMaster within reaching distance of the sensor (Fig. 2). The LCD screen was mounted on the wall in front of them; the foot switch was placed on the floor at their feet. The experimenter sat on the opposite side of a partition and observed participants via a

webcam. For the JR task, participants sat with their right elbow resting on a marked position on a desk. During this task, the experimenter sat next to them to read the angles of the forearm positions.

## Procedure

### *Dynamic movement reproduction task*

During the DMR task, participants replicated square (practice) and circular movement patterns while holding the sensor of the HapticMaster. Movement direction was counterbalanced: some participants moved in the clockwise direction during the entire Visual Information condition and counterclockwise during the entire No Visual Information condition (Fig. 1), other participants received the reversed combinations. Therefore, none of the findings can be attributed to specific combinations of stimuli. Each condition started with three practice trials to familiarize participants with the procedure, followed by six test trials.

#### *Practice phase*

Instructions on how to operate the HapticMaster and the procedure of the task were presented on-screen, including movement direction (and pattern shape; Fig. 1), and whether or not a blindfold would be worn. Additionally, participants were informed that all movements would occur in the horizontal plane. On each trial, the HapticMaster first restricted movement to a single trajectory to show participants what movement was to be reproduced (i.e., *target movement*). The participants actively explored the trajectory; the HapticMaster was programmed to haptically block certain areas of its workspace, thus restricting the movement to specific patterns. During practice, this pattern was a square with a side length of 16 cm. Note that on all practice trials the shape of this pattern was the same and starting positions were always in the middle of the side closest to the participant. Participants began movement when they heard a starting tone after the automated audio message "*Start guided movement*" (Fig. 2). If participants moved in the wrong direction, an error message was played and the trial restarted. After being guided through the target movement once, they were asked to reproduce this movement as accurately as possible, while having the entire range of the robotic arm—within the horizontal plane—available. Participants began moving when they heard a starting tone after the automated audio message "*Start free movement*", and participants verbally said, "*Stop*" when they finished reproducing the movement, at which time the experimenter manually ended the trial. The robotic arm then moved to a new starting position for the next trial. Six different starting positions within the same horizontal plane were used (random order; shoulder angles along the frontal plane approximately between 0 and 90 degrees; shoulder angles along the longitudinal plane approximately between 0 and 45 degrees; elbow angle approximately between 30 and 160 degrees) to limit potential spatial learning effects. No feedback regarding participants' performance was given.

#### *Test phase*

Identical procedures to the practice phase were used except the shape of the target movements was changed to a circle with a radius of 8 cm (to limit potential training

effects). Note that on all test trials the shape of this pattern was the same and the starting position was always on the point of the circular pattern closest to the participant.

### Joint reposition task

Each condition started with one practice trial to familiarize participants with the procedure, followed by six test trials. During the task, participants' elbow rested on a marked spot on a horizontal surface (shoulder angle along the frontal plane at approximately 0 degrees; elbow angle at approximately 135 degrees; wrist in neutral position, with thumb pointing upwards; see Fig. 1). Prompted verbally by the experimenter, participants actively moved their forearm to a target position, moved back to resting position on the horizontal surface, and then actively reproduced the target position (Fig. 3). Participants were allowed to adjust the position of the forearm until they felt it reached the correct position, and verbally indicated when this was the case. Both the target and reproduced angle were recorded by the experimenter. Three different target angles (30°, 45° and 60°; between forearm and horizontal surface) presented in random order were assessed twice. No feedback regarding participants' performance was given.

## Main outcome variables

### Dynamic movement reproduction error

DMR error was operationalized as the mean absolute difference (in cm) between the reproduced and the target circular movement pattern (i.e., radius) on each trial (see Supplementary Material S1 for an alternative measure considering direction of errors). Larger errors reflect poorer accuracy. The reproduced radius was calculated using the coordinates of each performed movement, as logged by the HapticMaster (Fig. 4).

### Joint reposition error

JR error is operationalized as the absolute difference between the target angle and the reproduced angle on each trial. Again, larger errors reflect poorer accuracy. The recorded target angles were approximately 30, 45 or 60 degrees—in most participants the recorded target angles were not exactly these values due to delay in stopping movement.

## Data preparation and statistical analysis overview

First, data were checked to determine if participants moved in the correct direction during the DMR task. This was assessed using the number of mistakes during guided movements, which was automatically logged by the HapticMaster. Next, data were visually inspected for other artifacts such as participants reaching the end of the movement plane of the HapticMaster.

The mean of each outcome variable was calculated per condition (over 6 trials) and is referred to as *accuracy*. The standard deviation of the measurements per condition is referred to as *consistency*, as it indicates whether subjects are consistent in their size/range of error (*Juul-Kristensen et al., 2008b*). Bland Altmann plots were used to visually inspect test–retest data of both tasks. Paired *t*-tests and Repeated Measures Analyses of Variance (RM ANOVAs) were calculated to check for systematic differences between sessions and trials. Intraclass Correlation Coefficients (*ICCs*; two-way mixed;

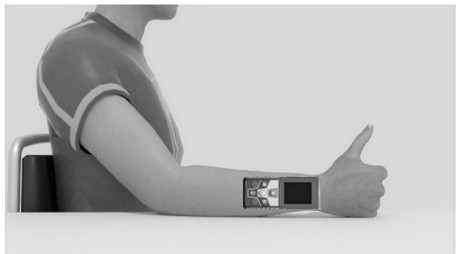

"Slowly, flex your arm, until I say stop"

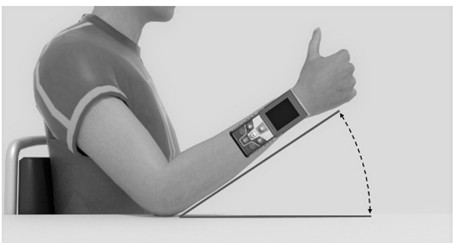

"Stop, hold and remember this position"
---Experimenter registers angle---
"Return your arm to the table and keep it there"

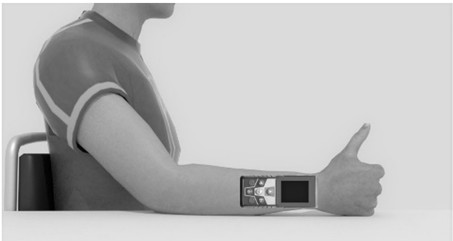

"Slowly flex your arm again to the same position as before and say, "Stop" when you think you reached it"

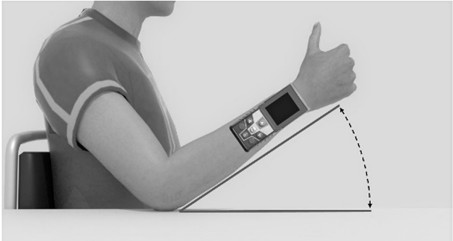

---Experimenter registers angle after participant says "Stop"---

**Figure 3** **Trial flowchart with experimenter's verbal instructions during Joint Reposition task.**

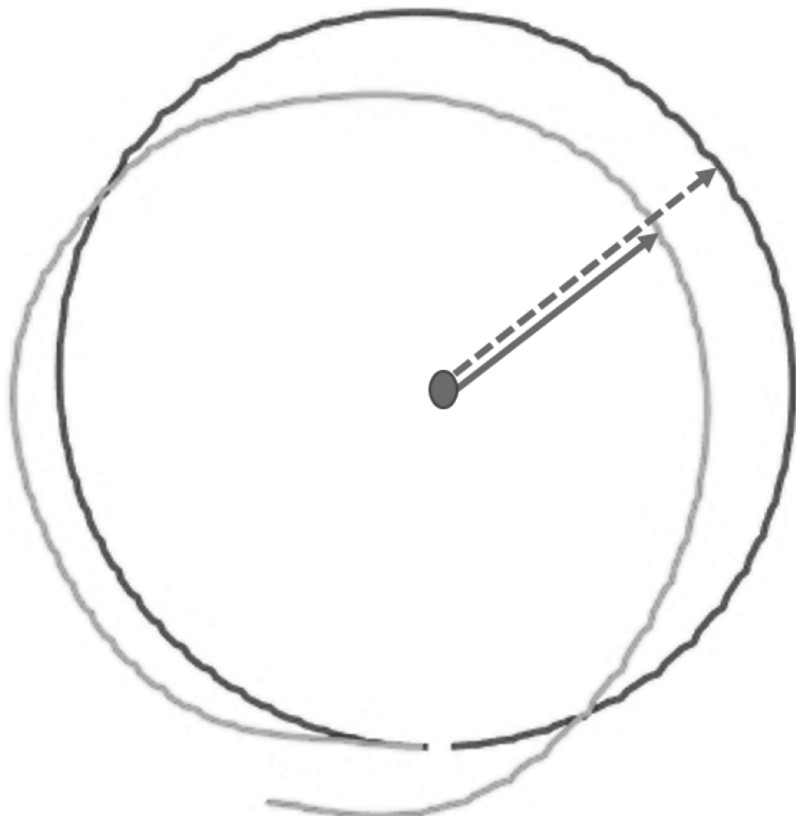

**Figure 4 Visualization of raw data from a single test trial of the Dynamic Movement Reproduction task.** Both the target (black line) and reproduced (gray line) movement pattern are visualized. Note that the dashed arrow represents the target radius, while the solid arrow represents the reproduced radius, which is calculated for each coordinate logged by the HapticMaster.

*McGraw & Wong, 1996*; *Shrout & Fleiss, 1979*) were used to test the absolute agreement between the test and retest sessions for accuracy and consistency (average measures). The categories of reliability used for reference were 0.00–0.40 (poor), 0.40–0.75 (fair-to-good), and 0.75–1.00 (good-to-excellent; *Fleiss, 1986*). We quantified measurement error of the DMR accuracy measure with the Smallest Real Difference (SRD; *Beckerman et al., 2001*). The SRD of a test is useful for both researchers and clinicians to determine whether a change in accuracy on the individual level is of significance at the 95% confidence level. First, the Standard Error of Measurement (SEM) was calculated using the standard deviation (SD) of all test–retest scores and the ICCs: $SEM = SD \times \sqrt{1-ICC}$ (*Chen et al., 2009*). Next, the SEM was used to calculate the SRD: $SRD = 1.96 \times SEM \times \sqrt{2}$. To evaluate if associations existed between task performance on both tasks, Spearman rank-order correlations, $\rho$, were calculated between DMR and JR accuracy, and consistency (DMR accuracy in Session 2, and JR accuracy and consistency in Session 1 were not normally distributed; Shapiro Wilk test $p < 0.05$). All correlations were calculated with and without outliers (>+3 $SD$ or <−3 $SD$). All analyses were performed on data from the no visual information condition, as this is the preferred way to test proprioceptive function. To test the

influence of visual information on task performance, RM ANOVAs were conducted on DMR and JR accuracy. The family-wise α was kept at 0.05. Bonferroni corrections were used to account for multiple testing. All statistical analyses were performed using SPSS 25 (IBM, Armonk, NY, USA). HapticMaster data was pre-processed using a MATLAB script (The MathWorks Inc., Natick, MA, USA).

## STUDY 1: RESULTS

### Test–retest reliability

*Dynamic movement reproduction accuracy (no visual information)*
Bland Altmann plots (Fig. 5) suggest that there is sufficient test–retest reliability for DMR accuracy and consistency. There was no statistically significant difference between sessions for accuracy, $t(22) = 1.54$, $p = 0.14$, nor consistency, $t(22) = −0.12$, $p = 0.90$. Adding trial as a factor in a RM ANOVA on DMR accuracy indicated no systematic differences on this level ($F < 1$). DMR accuracy had good-to-excellent reliability, $ICC = 0.80$, $F(22,22) = 5.35$, $p < 0.001$, 95% CI [0.55–0.92]. The SRD value for DMR accuracy is 0.76 cm. In other words, a change between two measurements of the same subject exceeding 0.76 cm can be interpreted as a true change at the 95% confidence level. Consistency showed fair-to-good reliability, $ICC = 0.63$, $F(22,22) = 2.64$, $p = 0.01$, 95% CI [0.11–0.85]. Sensitivity analyses without outliers yielded similar results.

*Joint reposition accuracy (no visual information)*
Bland Altmann plots (Fig. 6) suggest poor test–retest reliability for JR accuracy and consistency, as variation in means of sessions (i.e., between-subjects) was lower than variation in differences between sessions (i.e., within-subjects). No systematic differences between sessions were present; Accuracy: $t(22) = −0.61$, $p = 0.55$; Consistency: $t(22) = −0.01$, $p = 0.99$. Adding trial as a factor in a RM ANOVA on JR accuracy indicated no systematic differences on this level, $F(5,110) = 1.06$, $p = 0.40$. JR accuracy had fair-to-good reliability, $ICC = 0.46$, $F(22,22) = 1.83$, $p = 0.08$, 95% CI [−0.29–0.77], but this was not statistically significant (ICC did not significantly differ from zero). Analysis without outliers confirmed the poor reliability: the $ICC$ dropped to −0.76 ($F < 1$). JR consistency had poor reliability, $ICC = 0.33$, $F(22,22) = 1.46$, $p = 0.19$, 95% CI [−0.66–0.72] and was not statistically significant. Analysis without outliers confirmed the poor reliability for consistency, $ICC = −0.26$, $F < 1$.

### Association between performance accuracy on the dynamic movement reproduction and the joint reposition tasks (no visual information)

The Spearman correlations showed no relationship between DMR and JR accuracy during initial test sessions (Table 1). The retest sessions did show a significant positive correlation of moderate strength. This correlation remained significant when correcting for multiple testing, though no longer when conducting the analyses without outliers, $ρ = 0.32$, $p = 0.18$. Analyses of consistency yielded similar results.

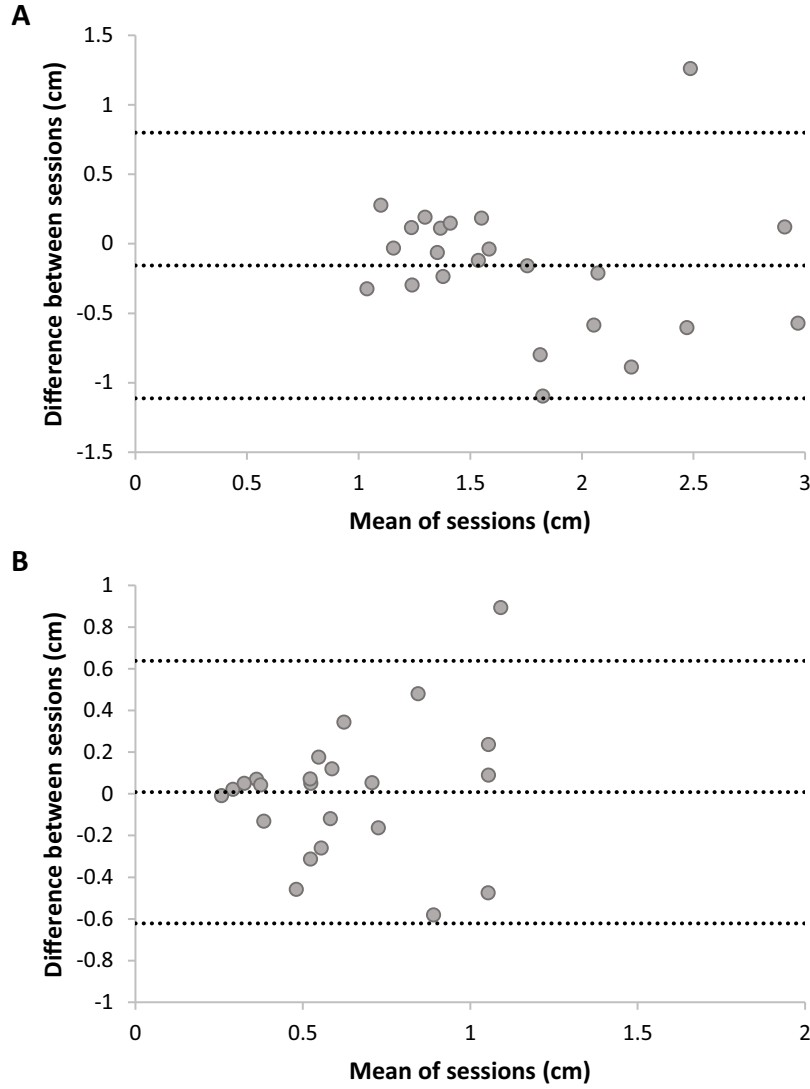

**Figure 5 Plots of test–retest data of Dynamic Movement Reproduction accuracy (A) and consistency (B) with no visual information.** The difference between Sessions 1 and 2 is plotted against the mean of both sessions. The mean difference between sessions is presented as a horizontal line (middle line), and the upper and lower lines represent the 95% upper and lower limits of these differences. Note that sufficient test–retest reliability corresponds with differences between sessions (*y*-axis) being closer to zero (i.e., roughly the same accuracy and consistency in both sessions), and variation in means between sessions (*x*-direction) being larger than variation in differences between sessions (*y*-direction; i.e., larger between-subjects variation than within subjects variation).

## The effect of visual information

### *Dynamic movement reproduction accuracy*

The 2 (Session: 1–2) × 2 (Visual Information: No visual information vs. Visual information) RM ANOVA analysis showed an effect of Session, $F(1,22) = 4.57$, $p = 0.04$, $\eta_p^2 = 0.17$, but no effect of Visual Information, $F < 1$, and no interaction, $F < 1$. Thus, visual information did not significantly influence DMR accuracy (Fig. 7). The significant effect

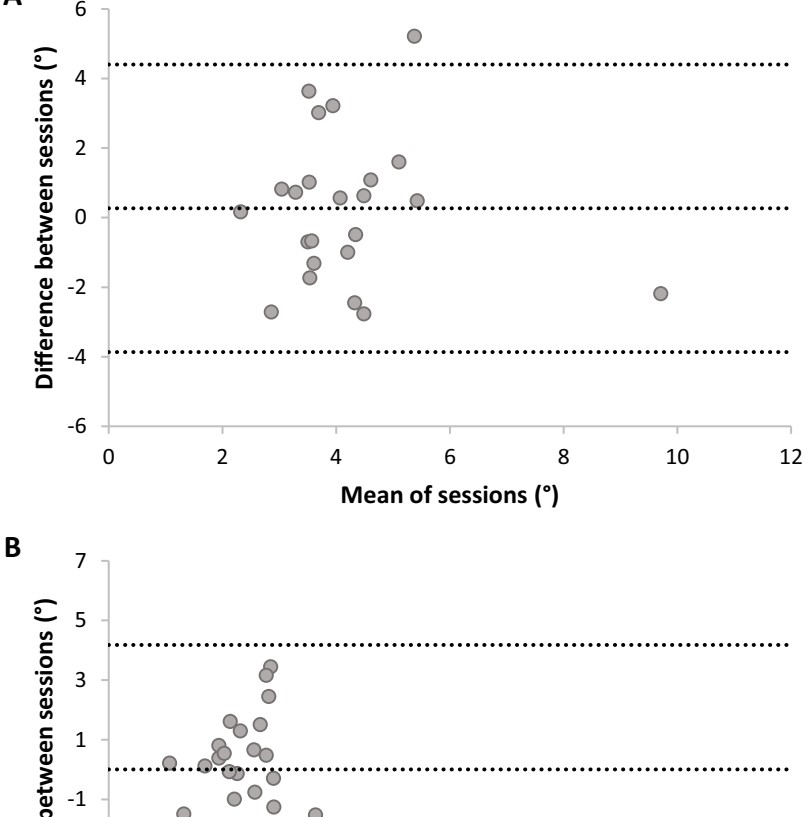

**Figure 6 Plots of test–retest data of Joint Reposition accuracy (A) and consistency (B) with no visual information.** The difference between Sessions 1 and 2 is plotted against the mean of sessions. The mean difference between sessions is presented as a horizontal line (middle line), and the upper and lower lines represent the 95% upper and lower limits of these differences. Note that sufficient test–retest reliability corresponds with differences between sessions (*y*-axis) being closer to zero (i.e., roughly the same accuracy and consistency in both sessions), and variation in means between sessions (*x*-direction) being larger than variation in differences between sessions (*y*-direction; i.e., larger between-subjects variation than within-subjects variation).

of Session suggests a potential learning effect (i.e., increased accuracy over sessions), as mean errors were significantly higher for Session 1 (*M* = 1.86; *SE* = 0.14) than Session 2 (*M* = 1.64; *SE* = 0.11).

### Joint reposition accuracy
A similar RM ANOVA showed an effect of Visual Information, $F(1,22) = 7.42$, $p = 0.01$, $\eta_p^2 = 0.25$, and no effect of Session, $F < 1$, or interaction, $F(1,22) = 1.12$, $p = 0.30$. Including

**Table 1 Spearman correlations (ρ) between Dynamic Movement Reproduction (DMR) and Joint Reposition (JR) accuracies, and consistencies (no visual information).**

| ρ (p-value) | | JR accuracy | | JR consistency | |
| --- | --- | --- | --- | --- | --- |
| | | Test | Retest | Test | Retest |
| Study 1 | | | | | |
| DMR accuracy | Test | 0.08 (0.72) | 0.39 (0.07) | | |
| | Retest | 0.05 (0.81) | 0.52* (0.01) | | |
| DMR consistency | Test | | | 0.06 (0.77) | 0.54* (0.01) |
| | Retest | | | 0.02 (0.93) | 0.27 (0.22) |
| Study 2 | | | | | |
| DMR accuracy | Test | 0.25* (0.05) | | | |
| DMR consistency | Test | | | 0.22 (0.08) | |

Note:
* Correlation is significant at the 0.05 level (2-tailed). No corrections for multiple testing.

visual information increased accuracy (i.e., smaller mean errors; No visual information: $M = 4.20$, $SE = 0.30$; Visual information: $M = 3.45$, $SE = 0.30$; Fig. 6).

## STUDY 2: MATERIALS & METHODS

The aim of the second study was to evaluate the reproducibility of findings regarding the association between both tasks and the influence of visual information using a larger, independent sample. Additionally, given the large sample, we considered the effect that task order might have on performance (first performing proprioceptive task with vision vs. first performing the task without vision). The apparatus, setting, procedure, and main outcome variables were identical to the first study, with the exception that this study was comprised of only one test session. In other words, participants performed both tasks only once, with the order of the tasks and conditions again randomized across participants.

### Participants

A convenience sample of 64 healthy volunteers was recruited (recruitment sources and eligibility criteria identical to Study 1; sample size based on power calculations for another—separately preregistered—research question). Statistical analyses were run on the complete sample of $N = 64$ (mean (SD) age = 22.33 (3.90), ranging from 18–37, 52 women). Participants received €12.5 in gift vouchers as compensation (part of a longer testing session).

### Data preparation and statistical analysis overview

Data were checked and prepared as described above. DMR consistency, and JR accuracy and consistency data were not normally distributed. Analyses and inference criteria were also identical, with the exception that paired $t$-tests were used to test for the effect of visual information. Additional RM ANOVAs were used to explore order effects (visual information vs. no visual information performed first).

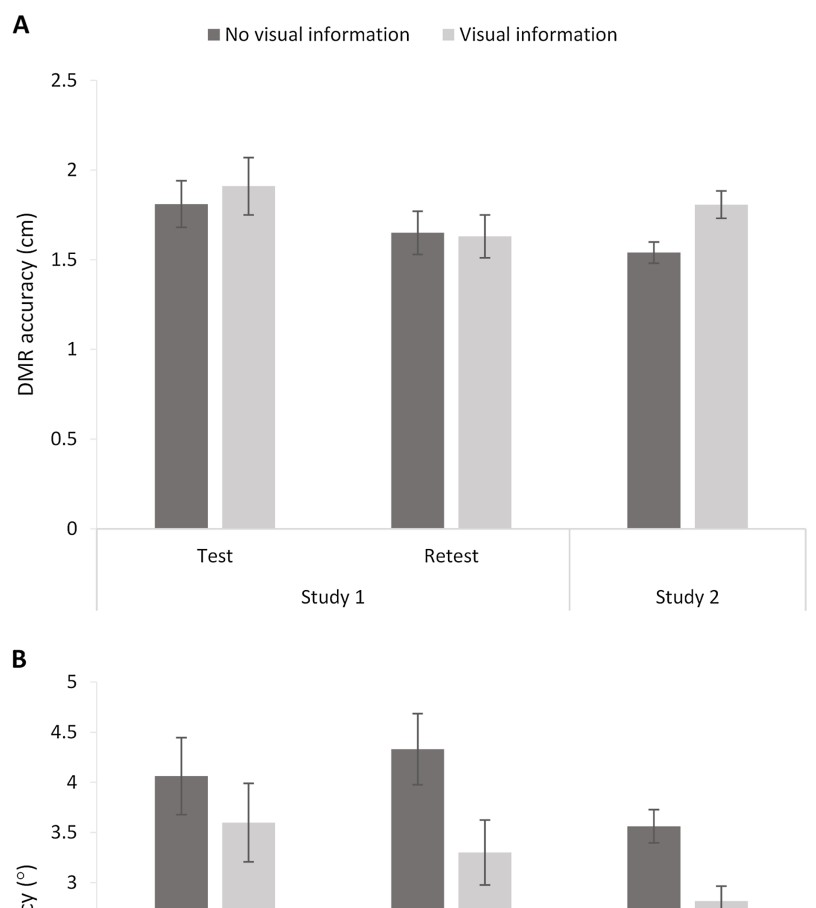

**Figure 7 Dynamic Movement Reproduction (DMR; A) and Joint Reposition (JR; B) accuracy with and without visual information during studies 1 (test and retest) and 2.** Higher values correspond with poorer accuracy. Error bars represent standard errors.

## STUDY 2: RESULTS

### Association between performance accuracy on the dynamic movement reproduction and joint reposition tasks (no visual information)

Spearman correlations suggest a weak positive relationship between DMR and JR accuracy (Table 1). Excluding outliers reduced the correlation further rendering it non-significant,

$\rho = 0.22$, $p = 0.09$. Analyses of consistency yielded comparable results (excluding outliers: $\rho = 0.17$, $p = 0.19$).

## The effect of visual information

### *Dynamic movement reproduction accuracy*

DMR accuracy differed significantly between conditions, $t(63) = -3.33$, $p < 0.005$, $d = 0.49$, showing higher errors when visual information was present ($M = 1.81$, $SE = 0.08$) than when no visual information was present ($M = 1.54$, $SE = 0.06$). These findings suggest poorer accuracy when vision was available. However, further exploratory analysis showed that condition order moderated the effect (Visual Information x Order: $F(1,62) = 7.16$, $p < 0.01$, $\eta_p^2 = 0.10$). More specifically, there was no significant difference between conditions when performing the task *without* visual information first ($p = 0.52$), but there was a difference between conditions when performing the task *with* visual information first ($p < 0.001$). In sum, visual information decreased accuracy, but only when the task started with visual information available.

### *Joint reposition accuracy*

JR accuracy differed significantly between conditions, $t(63) = 3.49$, $p < 0.001$, $d = 0.59$, with lower errors when visual information was present ($M = 02.82$, $SE = 0.15$) than when no visual information was present ($M = 3.56$, $SE = 0.17$). Thus, participants showed poorer accuracy without visual information available. Adding the order of conditions as a factor in a RM ANOVA did not change the results (Visual Information x Order: $F < 1$).

## DISCUSSION

We developed a new task to dynamically assess sensorimotor integration underlying kinesthetic proprioceptive function through measures of kinesthetic proprioceptive accuracy during upper limb movement, and evaluated test–retest reliability, association with a JR task, and the influence of visual input on task performance. Our first hypothesis, that the DMR task test–retest reliability would be good to excellent was supported, as was the presence of weak associations between the DMR task and the JR task. Interestingly, allowing the use of vision during the DMR task was different than we hypothesized— vision only improved task performance for the JR task, not the DMR task. Importantly, our latter findings on vision and task performance associations were largely reproducible in a larger independent sample, although the effect of vision on DMR task performance was not.

An important implication of the present study is that our new DMR task evaluating kinesthetic proprioceptive function exhibits sufficient test–retest reliability to support its use in research and clinical settings. That is, DMR accuracy showed good-to-excellent reliability, and fair-to-good test–retest reliability was found for consistency (i.e., variation in error throughout the task). These results are comparable to other studies evaluating test–retest reliability of tasks assessing proprioceptive function using robotic devices (e.g., *Cappello et al., 2015*; *Rinderknecht et al., 2018*; *Rinderknecht et al., 2016*). Furthermore, this reliability was notably higher than that of the traditional JR task tested

here. However, it is of interest that our findings of poor test–retest reliability for JR accuracy differed from that of *Juul-Kristensen et al. (2008b)*, who found fair-to-good reliability. There could be numerous reasons underlying these differences between our studies. While device precision appears comparable, it is possible that use of differing forearm repositioning angles (smaller angles in our task which may increase task ease, reducing between-subject variability and thus reliability), differing duration between test sessions (1 h vs. 24 h in present study, enhancing memory of the task and increased reliability in former work), participants' age range (18–57 years vs. 18–32 years in present study, reducing between-subject variability), and examiner experience (highly trained physiotherapists in past work, psychology student here) all contributed to differing findings. Together, our differing reliability findings for the JR task support past work showing that reliability varies widely (*Clark, Röijezon & Treleaven, 2015*; *Elangovan, Herrmann & Konczak, 2014*). Regardless, it is important to highlight that potential memory and/or examiner effects did not influence the DMR measure to the same extent, thus highlighting the benefit of using a highly standardized protocol and a device allowing for precise measurement such as the HapticMaster.

The second implication of this study is that the DMR task measure of proprioceptive accuracy captures a unique aspect of proprioceptive function; that is, distinct from the JR task. Indeed, the association between DMR and JR accuracy was poor. Importantly, we replicated this finding in our second study using a larger sample, collected to remove any concerns about low statistical power in Study 1. While both tasks involve active reproductions involving the elbow joint of the dominant (right) arm which might suggest stronger association, in general, measures of proprioceptive function correlate weakly, as they assess different aspects of proprioception (*De Jong et al., 2005*; *Elangovan, Herrmann & Konczak, 2014*). In the DMR task, participants perform a specific movement *pattern* (i.e., a circle) and then reproduce this pattern. This is similar to the JR task where participants move their arm to a certain *position* and then reproduce it. That is, the DMR task involves remembering the position and size of the circle and using (combined) limb position sense to replicate that circular movement, as well as sensorimotor integration to ensure accurate performance of the intended action. The latter integration is likely underpinned via generation of a motor efferent copy, which is then compared to sensory input that has resulted as a consequence of the movement (*Wolpert & Ghahramani, 2000*; *Wolpert et al., 1995*). However, the purposeful complex movements during the DMR task require higher levels of sensorimotor integration compared to the JR task. Furthermore, the error measure in the DMR task is not only looking at an 'end position' but the accuracy of the entire movement (i.e., dynamic assessment). Our findings suggest then that the DMR task mainly captures sensorimotor integration processes underlying kinesthetic proprioceptive function, rather than joint position sense. Furthermore, it is of course, relevant to note that the DMR task involves the entire upper limb kinetic chain (shoulder, elbow and wrist joints) while the JR task only involves elbow movement, which may also contribute to poor associations. Together these results emphasize the importance of considering and appropriately assessing the proprioceptive feature of interest as well as the relevant joints for that condition. Overall, our findings support our hypothesis that

the DMR task indeed captures a unique aspect of proprioceptive function compared to JR tasks.

Finally, our results suggest that the contribution of visual information to proprioceptive accuracy is complex. It appears dependent on the proprioceptive task, and for the DMR task, was opposite of what we expected. That is, while our first study found no difference between visual conditions when performing the DMR task, exploration of this in a larger sample showed lower accuracy when performing the task with visual information. Additionally, the condition order moderated this effect, as it was only present when participants started the task with visual information present. In contrast, JR results were consistent with expectations, indicating that visual information increased accuracy (*Scheidt et al., 2005*). These contradicting results of vision for the JR vs. DMR task may be explained by task differences. For example, in the JR task, simple forearm positions are reproduced (i.e., allowing the use of visual reference points), while in the DMR task more complex movements (including movement of a robotic arm) using multiple joints are performed, potentially making reliance upon visual information a disadvantage. Further, order effects for the DMR task may be explained by the influence of visual information while learning a task. For example, if visual information is absent while learning a complex task, it is learned proprioceptively; the visual information that is available afterwards is then an adjunct to the proprioceptively learned task. The same may not occur if visual information is present while learning a complex task—it may provide less reliable information for a complex movement than the proprioceptive input. This emphasizes that the integration of visual and proprioceptive information is not straightforward (*Sarlegna & Sainburg, 2009*; *Scheidt et al., 2005*), and supports that proprioceptive and visual information are weighted based on their reliability (*Van Beers, Sittig & Van der Gon, 1999*).

Some limitations of the current study should be outlined as well. First, analysis of the effect of visual information on the DMR task revealed the presence of a potential learning effect between sessions (i.e., higher accuracy in the second session), even though our initial analysis did not show this effect and test–retest reliability was in the good-to-excellent range. However, this may have led to an underestimation of test–retest reliability. Increasing practice or familiarization when using tasks that are more complex may be advisable. Second, we did not fully standardize movement kinematics of the upper limb during the DMR task, allowing some variation in use of different joints during movements (i.e., only movement of the sensor of the robotic arm was recorded). When comparing accuracy within or between participants, this should be standardized in order to prevent compensation for joint-specific deficits. Third, the sample size of our first study may not have provided sufficient statistical power for the JR test–retest analysis, although it was sufficient for the DMR measures. Fourth, since the average age of our sample is relatively low, the present findings are limited in generalizability toward older adults. Finally, as with all assessments of active proprioceptive function, factors other than proprioception can influence the outcomes. For example, both the DMR and the JR methods are less suitable for people with severe cognitive impairments since the tasks depend on working memory (*Han et al., 2016*). Another factor is motor control: the precision of movement limits the precision of the measured proprioceptive accuracy

(*Elangovan, Herrmann & Konczak, 2014*). However, motor control and proprioception are closely related as both are integrated to perform movements (*Proske & Gandevia, 2012*).

The DMR task's highly automated (i.e., requiring limited operator input) and brief (around 15 min) protocol makes it straightforward to use in a clinical setting. Our results indicate that a change of 0.76 cm or more in DMR accuracy (i.e., the difference between two measurements) is meaningful, and not merely due to measurement error (*Beckerman et al., 2001*), highlighting its precision. It should be noted that the task presented here is not necessarily limited to use of HapticMaster devices, as similar devices, commonly found in research and rehabilitation settings (e.g., motor cortex retraining in stroke patients; *Timmermans et al., 2009*), could be used. Further, our DMR task using a robotic device, while not the first, extends previous work in this area. For example, in contrast to our DMR task, past active robotic tasks have depended on visual guidance (*Dukelow et al., 2012*), or used mirror-matching (*Kenzie et al., 2014*). Additionally, *Kitchen & Miall (2019)* have used a robotic device in older adults to evaluate arm-reaching movements, though their task involves reaching a position along a straight line as quickly as possible, and thus does not require complex sensorimotor integration. Rather, the DMR is the first to use vision occluded complex movements, particularly emphasizing sensory guidance by using replication of a circular pattern and not emphasizing speed, making it more appropriate to assess kinesthetic proprioceptive function and the sensorimotor integration processes required for accurate arm movement pattern reproduction.

The innovative aspect of our DMR task is that it dynamically assesses sensorimotor integration processes of the entire upper limb kinetic chain, potentially allowing capture of more complex kinesthetic proprioceptive deficits. In addition to the device's precise measurement, this could help capture deficits in certain conditions that involve multi-joint impairment more accurately, which is of importance in both research and practice (*Röijezon, Clark & Treleaven, 2015*; *Stasinopoulos, 2019*). For example, lateral epicondylalgia (LE), or 'tennis elbow' is characterized by symptoms of persistent pain and sensorimotor dysfunction, and people with LE present with impaired proprioceptive function at the elbow (*Juul-Kristensen et al., 2008a*). However, recent work has shown that sensorimotor dysfunction also occurs at the shoulder, the scapula, and the wrist (*Alizadehkhaiyat et al., 2007*; *Day et al., 2015*; *Lucado et al., 2012*), and that it may be the combination of impairment within this dynamic upper extremity kinetic chain that impedes treatment (*Lucado, Vincent & Day, 2019*). However, reliable, valid proprioceptive tests to evaluate the entirety of the upper limb kinetic chain are currently lacking, limiting detection of impairment and provision of appropriate treatment (*Stasinopoulos, 2019*). Future research is warranted to explore use of the DMR task in clinical populations such as LE, evaluating test–retest reliability as well as exploring the predictive validity of the DMR measure for clinical improvement via proprioceptive or movement retraining. Visual inspection of the reproduced (vs. target) movement pattern data may also prove useful in clinical application, as it may allow identification of what aspect of proprioceptive function to target (see Supplementary Material S1 for examples of impaired position sense, but intact sensorimotor integration and vice versa). Further work exploring combination of the

DMR task with movement capture systems to explore differences in the way movement occurs during the task might provide interesting insight.

## CONCLUSIONS

In conclusion, the DMR task seems a promising new tool for reliably testing kinesthetic proprioceptive function of the upper limb. It showed high test–retest reliability, and appears to capture a unique aspect of proprioceptive function, as it dynamically evaluates sensorimotor integration processes of the entire upper limb. This may make the DMR task particularly relevant for certain clinical conditions with multiple-joint involvement. Additionally, this study shows that the integration of visual and proprioceptive information is not straightforward, with vision of arm movement beneficial during simple movements, but not when learning complex movements, and supports the idea that proprioceptive and visual information are weighted based on their task-specific reliability.

## ACKNOWLEDGEMENTS

The authors wish to thank Jacco Ronner for programming the DMR task and Vanessa Lostaunau Calero for assisting in data collection.

### Funding

This research is supported by a Vidi grant from the Netherlands Organization for Scientific Research (NWO), The Netherlands (grant ID 452-17-002) granted to Ann Meulders. Tasha Stanton is supported by a National Health & Medical Research Council of Australia Career Development Fellowship (ID1141735). The funders had no role in study design, data collection and analysis, decision to publish, or preparation of the manuscript.

### Grant Disclosures

The following grant information was disclosed by the authors:
Netherlands Organization for Scientific Research (NWO), The Netherlands: ID 452-17-002.
National Health & Medical Research Council of Australia Career Development Fellowship: ID1141735.

### Competing Interests

The authors declare that they have no competing interests.

### Author Contributions

- Kristof Vandael conceived and designed the experiments, performed the experiments, analyzed the data, prepared figures and/or tables, authored or reviewed drafts of the paper, and approved the final draft.
- Tasha R. Stanton conceived and designed the experiments, authored or reviewed drafts of the paper, and approved the final draft.

- Ann Meulders conceived and designed the experiments, authored or reviewed drafts of the paper, and approved the final draft.

## Human Ethics

The following information was supplied relating to ethical approvals (i.e., approving body and any reference numbers):

The experimental protocol was approved by the Ethics Review Committee Psychology and Neuroscience of Maastricht University (185 09 11 2017 S5).

## Data Availability

The raw data files, final datasets, a document describing the variables contained in the raw data files of the DMR task, and the scripts to calculate mean absolute deviations and plot movements of the DMR task are available in the Supplemental Files.

## Supplemental Information

Supplemental information for this article can be found online at http://dx.doi.org/10.7717/peerj.11301#supplemental-information.

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
