# Peer review of "Assessing kinesthetic proprioceptive function of the upper limb: a novel dynamic movement reproduction task using a robotic arm"

_PeerJ, doi:10.7717/peerj.11301_

## Round 0.1 · original submission · Major Revisions

Dear Authors,

Your review has now been completed. You will notice that we had two detailed and comprehensive reviews of your work and I believe that these comments can help improve the quality of this manuscript.

While the reviewers believed that the manuscript was interesting and the study relevant, you will find a general consensus that more details are required. Reviewer comments can be found attached and listed below. We look forward to your reply.

Regards,

Reviewer 1 ·

Basic reporting

No comment

Experimental design

Methods should be improved.

Validity of the findings

No comments.

Annotated reviews are not available for download in order to protect the identity of reviewers who chose to remain anonymous.

Reviewer 2 ·

Basic reporting

The paper is well written and easy to read. The structure with the two studies being presented separately is not ideal (creates some repetition, and many subsections), but this is a rather minor point.
In the introduction and discussion, authors should better define/justify the aspect of proprioception they target with their task, and consider further comparison with other work that looked at proprioception assessments using robotic platforms (please refer to the general comments section for more details).
Finally, a figure presenting raw data (e.g., from one or two subjects) would be helpful to better understand how the outcome measure of the DMR task are calculated.

Experimental design

In general, the experiments and the data analysis seem suitable. Additional details on the procedure should be clarified or better motivated (please refer to the general comments section for more details).

Validity of the findings

Providing further data on measurement errors (smallest real difference) would help better evaluate the task and its potential for clinical applications. Given the different nature of the two tested tasks, their direct comparison (and the authors hypotheses in respect to that) is somewhat puzzling. The effect of visual information is a very interesting finding of the paper, however there is no strong conclusion about the actual mechanisms causing the observe differences between tasks. Please refer to the general comments section for more details.

Additional comments

In this paper, authors present a novel assessment of proprioceptive function (the DMR task) consisting of the reproduction of arm movements via a robotic platform, investigate its reliability and compare it to a joint reposition (JR) task. Furthermore, the role of visual information was investigated in both tasks. The paper is well written and the research is of interest (especially the part on the role of visual information). The motivation for the proposed task and its novelty/uniqueness compared to other such robotic assessments could be better highlighted.

Authors should consider the following points to improve their manuscripts.

1) Proprioception is complex and authors should try to better differentiate between the different aspects of proprioception, which contains position sense and kinaesthesia. It seems that the reproduction of dynamic movement patterns would be a kinaesthetic task (e.g., focusing on the perception of motion) while a test like the Joint Repositioning task is a measure of position sense. Thereby, it would not be surprising that the 2 are only poorly associated, since addressing different aspects of proprioception. This should be better discussed (potentially discussing the physiological process involved in the two), and the terminology might need to be adapted. For example, statements such as (in the conclusion) “the DMR task is promising to reliably test proprioceptive function” might need to be adapted to specifically mention kinaesthesia. Also, the hypotheses/expectations of the authors on that matter in the introduction (line 112-116) are rather confusing.

2) Addressing reliability of proprioception measures using robotic devices, authors should compare their work and results to other research that investigated this in healthy subjects and neurological patients. See for example the work of:
Rinderknecht et al. 2016 doi.org/10.3389/fnhum.2016.00316
Rinderknecht et al. 2018 doi: 10.1186/s12984-018-0387-6
Capello et al. 2015 doi:10.3389/fnhum.2015.00198
It would also be interesting to compare the task and its outcomes in terms of clinical applicability (e.g., time for the assessment, space required by the robot, amount of information that can be gathered, etc).

3) Regarding the procedure, it would be important to further explain a few methodological points and the rationale behind them. In particular:
-it is unclear whether the reference movement on the HapticMaster is performed passively by the robot (i.e., the robot moves the subject) or whether the subject has to actively explore the shape while the robot “restricts” the movement (if so, how is this done?).
-as authors briefly mention in their discussion, the shapes presented in the DMR task are complex and drawing them in a plane requires the combination of all joints of the upper limb (and there would be multiple possible joint configurations that would lead to an accurate movement reproduction). It is unclear how such a task could directly be compared to the JR task where one single joint is probed. While it can be argued that the DMR task probes proprioceptive function (e.g., kinaesthesia), it certainly does it in a very different way than JR, and this should be further highlighted. The fact that the DMR task involves multiple joints should be further discussed (possibly also as a reason for visual information not helping, since potentially less helpful or even unreliable for such complex movements?).
-Please specify what vision means. Is there visual feedback provided on a screen (e.g., information about end-point position in real time?) or is this simply that subjects can see the movement of their arm?
-It is unclear why, in the JR task, the experimenter did not physically move the arm to the target angle during the presentation of the reference position. The procedure with verbally indicating subjects to stop seems very inefficient and prone to generate errors and variability. Also, why was only one practice trial (vs 3 in the DMR) provided? This methodological decision may also influence the observed variability in the JR task.

4) Regarding the outcome variables, it is really unclear how the DMR error is calculated (is this based on a fit? is this calculated in polar coordinates?). A figure representing examples of raw data would be very helpful to visualise this metric.

5) In the discussion, authors touch upon clinical usability of their novel task. Besides reliability and ICC values, it would be very important and indicative to report measurement errors (in terms of smallest real difference) allowing to put into perspective the variability of one subject between sessions with the actual range of values (variability) among all subjects. This should help understand what magnitude of a change or deviation could be understood as (clinically) relevant when applying such test longitudinally.

6) When comparing reliability of the JR task to the work of Juul-Kristensen et al. 2008b (from line 361), one additional potential difference to discuss might be the number of subjects and their age distribution. It is known that proprioception declines with age, and since they seem to have tested a wider age range, they might have probed a less homogeneous population, which is in general beneficial for a reliability analysis. In general, the numbest of subjects in the first part of the present study seem rather small for a reliability analysis, especially in an homogeneous group of healthy subjects. This should be listed as a limitation. Also, it would have been nice to test subjects of older age.

Minor comments:
-Please consider adapting the title, as “using robotic arm movements” is very vague. It would be more informative to underline what makes this task novel.

-Abstract- Results: Please be specific and directly mention visual information and not “sensory input” in the fourth sentence, since authors specifically tested for the addition of visual information and not any sensory input.

-Abstract - Discussion: In the first sentence (and in other occurrences in the paper), it is not clear why authors refer to kinetic chain. Should this not be kinematic chain?

-line 65-69: It seems rather obvious that assessment quantifying position sense (i.e., once a posture is reached) provide little insight on dynamic movements, since these tasks should be measuring how well a static arm configuration is reproduced. Please consider rephrasing.

---

## Round 0.2 · accepted · Accept

Congratulations on the acceptance of this work. Thank you for addressing our reviewer comments, the revisions were very well done.

Reviewer 1 ·

Basic reporting

Is good and clear.

Experimental design

Is great.

Validity of the findings

Is acceptable.

Additional comments

I would like to thank the authors for addressing my initial comments. Following the revision to the article, the current version is a clearer, concise, and well-written manuscript.